# How Heat Transfer Indirectly Affects Performance of Algae-Bacteria Raceways

**DOI:** 10.3390/microorganisms10081515

**Published:** 2022-07-26

**Authors:** Francesca Casagli, Olivier Bernard

**Affiliations:** 1Biocore, Inria Centre at Université Côte d’Azur, INRAE, 2004 Route des Lucioles, 06902 Sophia-Antipolis, France; olivier.bernard@inria.fr; 2LOV (Laboratoire d’Océanographie de Villefranche), Sorbonne Université, CNRS UMR 7093, 181 Chem. du Lazaret, 06230 Villefranche-sur-Mer, France

**Keywords:** microalgae, bacteria, modelling, wastewater treatment, liquid depth, temperature

## Abstract

Oxygenation in wastewater treatment leads to a high energy demand. High-rate algal-bacterial ponds (HRABP) have often been considered an interesting solution to reduce this energy cost, as the oxygen is provided by microalgae during photosynthesis. These complex dynamic processes are subject to solar fluxes and consequently permanent fluctuations in light and temperature. The process efficiency therefore highly depends on the location and the period of the year. In addition, the temperature response can be strongly affected by the process configuration (set-up, water depth). Raised pilot-scale raceways are typically used in experimental campaigns, while raceways lying on the ground are the standard reactor configuration for industrial-scale applications. It is therefore important to assess what the consequences are for the temperature patterns of the different reactor configurations and the water levels. The long-term validated algae-bacteria (ALBA) model was used to represent algae-bacteria dynamics in HRABPs. The model was previously validated over 600 days of outdoor measurements, at two different locations and for the four seasons. However, the first version of the model, like all the existing algae-bacteria models, was not fully predictive, since, to be run, it required the measurement of water temperature. The ALBA model was therefore updated, coupling it with a physical model that predicts the temperature evolution in the HRABP. A heat transfer model was developed, and it was able to accurately predict the temperature during the year (with a standard error of 1.5 ${}^{{}^{\circ}}$C). The full predictive model, using the temperature predictions, degraded the model’s predictive performances by less than 3%. N${}_{2}$O predictions were affected by ±7%, highlighting the sensitivity of nitrification to temperature The temperature response for two different process configurations were then compared. The biological process can be subjected to different temperature dynamics, with more extreme temperature events when the raceway does not lie on the ground and for thinner depths. Such a situation is more likely to lead to culture crashes.

## 1. Introduction

In the last decades, microalgae have appeared as a new promising biological resource and received much attention. The market for valuable products derived from microalgae is growing rapidly [1], with applications in fish farming, bio-plastics, the food industry, and cosmetics [2]. Biomass production is mainly carried out in open raceways [3]. These outdoor paddle-wheel-agitated tanks are cheap production systems, and there has been an intense ongoing debate as to whether their actual productivity compared to closed photobioreactors is significantly lower [4]. Microalgae cultivated in raceways in association with bacteria are also considered as an interesting solution to reduce the energy demand for oxygenation in wastewater treatment, since oxygen can be provided by the microalgae during photosynthesis.

The performance of these systems is first determined by the amount of photons received by the microalgae, which triggers the photosynthetic reactions. Temperature is the second factor affecting the growth rate of microalgae [5], and it has received much less attention. In fact, temperature strongly affects system performance, especially in outdoor systems, where it is subject to large fluctuations both during the day and seasonally. Predicting the productivity that a raceway can achieve at a given location and for a certain period of the year is becoming a key issue for retrofitting current wastewater treatment plants, and for including microalgae in the process. For this, on top of a biological model, a model predicting the water temperature is necessary.

It turns out that the process configuration (set-up, water depth) can strongly drive the temperature response. Since many studies at pilot scale consider raised raceways, the question that we target is how far will an up-scaled process made of concrete raceways lying on the ground differ from this behaviour due to a different temperature pattern.

Several models were developed to represent the dynamics of algal-bacterial processes, in particular in the field of wastewater treatment using high-rate algal-bacterial ponds (HRABP) [6,7,8,9,10,11]. Among these models, the originality of the ALBA model [10,12] is that it relies on an unprecedented validation of more than 623 days, carried out under outdoor conditions at two different locations and for the four seasons. However, until now, none of these HRABP models were fully predictive, as they require knowledge of the water temperature (or of the temperature below the raceway).

The first objective of this work is to propose a heat transfer model to represent the temperature evolution in one of the outdoor pilot raceways used for validating the ALBA model. This pilot scale experimental raceway (3.8 m${}^{2}$), located in northern Italy, was used to process digestate, and six months of records were used to validate the biological model. The raceway was typical of pilot set-ups for investigating and assessing the process performances in realistic conditions.

Several heat transfer models have been proposed for algal raceways. The most popular one was developed by Béchet et al. [13,14]. A simplified model was proposed by Slegers et al. [15]. A combination of these two approaches was then used in Rodríguez-Miranda et al. [16] for a raceway pond. In this paper, we are focusing on a different configuration, where the raceway is not directly in contact with the ground. For this specific configuration, there is no heat transfer model available.

The heat transfer temperature model that we propose was modified from the work of Béchet et al. [13], by adapting it to the specific material and configuration of the reactor. In particular, the temperature dynamics for the raceway material were added, including the exchange with the liquid medium and the environment. This fully predictive version of the ALBA model is therefore able to predict the temperature and the variable volume of the raceway, as well as the dynamics of the process, including microalgae, heterotrophic bacteria, and nitrifying bacteria.

It is important to stress that there are many pilot-scale raceways that are not in direct contact with the ground; thus, their thermal inertia and the associated temperature dynamics differ from the ones at industrial scale. In fact, different reactor configurations are subjected to different heat-transfer contributions and balances, resulting in a different temperature inside the reactor. For instance, when the raceway is lying on the ground, the conduction from the ground mitigates its temperature, while when the reactor is raised up from the ground, the radiation from the ground heats the pond bottom.

With this new fully predictive model version, it became possible to compare the performances of a raised raceway with those of an industrial configuration, where the raceway is made of concrete and lies on the ground. Different water depths were also tested, and they corresponded to different levels of thermal inertia. Algae productivity, nutrient removal rates, and the amount of water lost by evaporation were assessed for these conditions. We show and quantify how the specific reactor design leads to different temperature dynamics, focusing on the extreme temperature values that can be reached in each configuration, especially in the warmest seasons, given their potential strong impact on the biological activity.

## 2. Materials and Methods

The HRABP was a pilot-scale polypropylene tank with an operating volume of 0.88 m${}^{3}$ and a total surface of approximately 3.8 m${}^{2}$ (see Figure 1). It was installed in the dedicated treatment plant of an intensive piggery farm, located in the north of Italy (Casaletto di Sopra, Cremona, Italy). The monitoring campaign was performed from May to December 2016. The HRABP had two main channels and semi-circular edges. The influent was prepared from the liquid fraction of the digestate from the biogas plant, and by dilution with tap water in order to reduce the influent TAN (total ammoniacal nitrogen) concentration. The influent characteristics were checked once per week, measuring the total suspended solids (TSS), the organic matter (COD total and soluble), phosphates ($P-PO_{4}^{3-}$), total ammoniacal nitrogen (TAN), total Kjeldahl nitrogen (TKN), and nitrate ($NO_{3}^{-}$). The effluent characteristics and therefore the reactor remediation efficiency were quantified by measuring: (i) once per week, the total (TSS) and volatile (VSS) suspended solids and the optical density at 680 nm; and (ii) twice per week with spectrophotometric test kits, the total and soluble organic matter ($COD_{T}$ and $COD_{S}$, respectively), the phosphates ($P-PO_{4}^{3-}$), and the inorganic nitrogen content ($NH_{4}^{+}$, $NO_{2}^{-}$ and $NO_{3}^{-}$). The HRABP was equipped with pH, temperature, and dissolved oxygen concentration probes. The reactor was also equipped with a contact cylinder (height 0.8 m, diameter 0.44 m, volume 0.12 m${}^{3}$) for bubbling pure CO${}_{2}$ gas to regulate pH at 7.5. The open pond mixing was ensured by a paddle wheel, which was operated at 20 rpm to obtain an average liquid velocity of 0.2 m s${}^{-1}$. The outflow was from a gravity overflow, setting a maximum liquid depth of 0.2 m and resulting in a variable volume according to the evaporation and rain contribution. An average hydraulic retention time (HRT) of 10 days was set until 11 October 2016; then, the HRT was set to 20 days.

Increasing the HRT is a typical operational strategy applied in outdoor bio-processes for wastewater remediation, to compensate for the lower temperature reducing microbial growth rates (typically in the coldest seasons, autumn and winter). This adaptation is necessary to match the limitations imposed by the national normative standards concerning nutrient discharge in water bodies. More details about the experimental campaign and the influent characteristics can be found in Casagli et al. [12], in Pizzera et al. [17], and in Supporting Information (SI.8).

### Methodology for Computing the System Key Performance Indicators

In order to assess the HRABP performances according to the two reactor configurations and the different liquid depths, the algal biomass productivity ($ALG_{prod}$), the total ammoniacal nitrogen removal rate ($TAN_{RR}$), and the nitrogen-to-algae yield ($ALG_{N,yield}$) were used as key performance indicators and computed as follows:(1)$$ALG_{prod}=\frac{X_{ALG}\cdot Q_{out}\cdot \phi }{S}$$
(2)$$TAN_{RR}=\frac{Q_{in}\cdot TAN_{in}-Q_{out}\cdot TAN_{out}}{S}$$
(3)$$ALG_{N,yield}=\frac{ALG_{prod}}{Q_{in}\cdot TAN_{in}}$$
where $X_{ALG}$ is the algae concentration in the reactor (gCOD m${}^{-3})$; *S* is the reactor surface (m${}^{-2}$); ϕ is the conversion factor from COD to algal dry weight $\left(0.64\;gDW\;gCOD^{-1}\right)$; $TAN_{in}$ and $TAN_{out}$ are the ammoniacal nitrogen concentration entering and leaving the system, respectively (gN m${}^{-3})$; and $Q_{in}$ and $Q_{out}$ are the inflow and outflow rate, respectively (m${}^{3}$ d${}^{-1}$).

## 3. Modelling Approach

### 3.1. The ALBA Model: Brief Recall of the Core Structure

#### 3.1.1. General Overview

The biological ALBA model considers a mixed culture of algae (X${}_{ALG}$), heterotrophic bacteria (X${}_{H}$), ammonium oxidizing bacteria (X${}_{AOB}$), and nitrite oxidizing bacteria (X${}_{NOB}$).

The model includes 19 biological processes and involves 17 state variables. The general model structure follows the general mass balance structure [18]:(4)$$\dot{\xi }=K\cdot \rho \left(\xi \right)-\Delta_{g}\left(\xi \right)+\frac{Q_{in}}{V}\cdot \xi_{in}-\frac{Q_{out}}{V}\cdot \xi$$
where $Q_{out}=Q_{in}+Q_{rain}-Q_{evap}$. ξ is the state variable vector (g m${}^{-3}$); $\xi_{in}$ is the vector of influent concentrations (g m${}^{-3}$); ρ(ξ) is the vector containing the reaction rates (g m${}^{-3}$d${}^{-1}$); $Q_{in}$, $Q_{rain}$, and $Q_{evap}$ are the inflow rate, rain rate, and evaporation rate, respectively (m${}^{3}$ d${}^{-1}$); *K* is the stoichiometric matrix (the transpose of the Peterson matrix); and *V* is the liquid volume (m${}^{3}$). The flux of gases dissolving or stripping from the liquid medium is denoted as $\Delta_{g}\left(\xi \right)$ (g m${}^{-3}$ d${}^{-1}$).

Hereafter, we recall the main structure of the ALBA model, but we refer the reader to the work of Casagli et al. [10,12] for more details.

#### 3.1.2. General Kinetic Model

The biokinetic rates are based on Liebig’s minimum law [19] for limiting substrates (carbon, nitrogen, and phosphorus), meaning that the most limiting nutrient drives the overall kinetics. The general expression describing the structure of the bioprocesses’ rates is given by:(5)$$\rho_{j}=\mu_{max_{j}}\cdot f_{T}\cdot f_{pH}\cdot f_{I}\cdot \frac{K_{n}}{K_{n}+S_{n}}\cdot \min_{i}\left(\frac{S_{i}}{S_{i}+K_{S_{i}}}\right)\cdot X_{BM_{i}}$$
where $X_{BM,i}$ is the biomass associated to the process $\rho_{j}$. The associated maximum specific growth rate is $\mu_{max_{j}}$ (d${}^{-1}$); $f_{pH}$ and $f_{I}$ are the functions describing pH and light dependence, respectively. The function $f_{T}$ represents the effect of temperature ($T_{p}$ being the pond temperature). The inhibition constant for the inhibiting substrate $S_{n}$ is $K_{n}$, and $K_{S_{i}}$ is the half-saturation constant for the limiting substrate $S_{i}$.

#### 3.1.3. Light Impact

Light is a crucial factor for algal growth, driving a large fraction of the energy and carbon fluxes in the system. The photosynthesis response to irradiance at depth *z* accounts for the photoinhibition phenomena at high irradiance with the Haldane-type function [20]:(6)$$f_{opt}\left(I\left(z\right)\right)=\frac{I(z)}{I\left(z\right)+\frac{\mu_{max}}{\alpha }{\left(\frac{I(z)}{I_{opt}}-1\right)}^{2}}$$

Light extinction is described by the Lambert–Beer equation:(7)$$I\left(I_{0},z\right)=I_{0}e^{-\epsilon X_{ALG}z}$$

Finally, the average growth rate for a given incident light and liquid depth is obtained, integrating the growth rate along the depth:(8)$$f_{I}\left(I_{0},h\right)=\frac{1}{h}\int_{0}^{h}f_{opt}\left(I\left(I_{0},z\right)\right)dz$$

#### 3.1.4. Temperature Impact

The temperature $T_{p}$ in the HRABP is generally not controlled, so it fluctuates within large ranges, driven by the solar heat flux, according to daily and seasonal dynamics. In the full predictive model, temperature is computed with a dedicated model as explained in the next paragraph.

The temperature dependence for algae and bacteria growth is modeled through the cardinal temperature model with inflection (CTMI), which requires three parameters, that is, the cardinal temperatures: $T_{max}$, $T_{min}$, and $T_{opt}$ [21]. The CTMI is zero for $T_{p}\notin \left[T_{min},\;T_{max}\right]$, where $T_{min}$ is the temperature below which no growth occurs and $T_{max}$ is the maximum temperature above which no growth occurs. $T_{opt}$ is the temperature at which the growth rate is maximum. For $T_{p}\in \left[T_{min},\;T_{max}\right]$, the temperature effect is given by the function $f_{T_{p}}^{1}$:(9)$$f_{T_{p}}^{1}=\frac{\left(T_{p}-T_{max}\right)\cdot {\left(T_{p}-T_{min}\right)}^{2}}{\left(T_{opt}-T_{min}\right)\cdot \Phi \left(T_{p}\right)}$$
where $\Phi \left(T_{p}\right)=\left(T_{opt}-T_{min}\right)\cdot \left(T_{p}-T_{opt}\right)-\left(T_{opt}-T_{max}\right)\cdot \left(T_{opt}+T_{min}-2T_{p}\right)$.

The Arrhenius function was chosen for the decay rates:(10)$$f_{T}^{2}=\theta {}^{T_{p}-20}$$
where the parameter θ describes the decay rate change with temperature.

#### 3.1.5. pH Impact

The pH strongly influences system dynamics, since it directly affects the dissociation of the majority of soluble compounds (S${}_{IC}$, S${}_{NH}$, S${}_{NO_{2}}$, S${}_{NO_{3}}$, S${}_{PO_{4}}$). The influence of pH on algae and bacteria bio-process rates is included through the function proposed by Rosso et al. [21], that is, the cardinal pH model (CPM). The CPM is zero for $pH\notin (pH_{min},\;pH_{max})$, where $pH_{min}$ and $pH_{max}$ are the minimum and the maximum pH thresholds, respectively. $pH_{opt}$ is the pH value at which the growth rate is maximum. For pH $\in (pH_{min},\;pH_{max})$, the pH effect is described by $f_{pH}$:(11)$$f_{pH}=\frac{\left(pH-pH_{max}\right)\cdot \left(pH-pH_{min}\right)}{\left(pH-pH_{min}\right)\cdot \left(pH-pH_{max}\right)-{\left(pH-pH_{opt}\right)}^{2}}$$

The pH is estimated by the model, resulting from the dynamical balance between the chemical, physical, and biological process interactions, as explained later on.

#### 3.1.6. Model Validation with Measured Medium Temperature

The model prediction capability was assessed on two case studies. A 56 m${}^{2}$ HRABP located in the south of France, with 1.5 year monitoring of pH, S${}_{O_{2}}$, S${}_{NH}$, S${}_{NO_{3}}$, S${}_{NO_{2}}$, X${}_{ALG}$, TSS, and soluble COD, was used to calibrate the model [10].

It was then successfully validated [12] on the case study presented in Section 2, where measured temperature was taken as input. Here, the objective is to predict temperature and to assess the prediction accuracy of the model coupling thermal and biological representations.

### 3.2. Predicting Pond Temperature and Water Level

The universal temperature model for shallow ponds developed by Béchet et al. [13] was adapted to predict the water temperature and the water level of the raceway. The existing radiative transfer model was modified, to deal with the fact that this raceway was suspended above the ground. A heat balance of the raceway material was thus also considered, involving additional heat fluxes (see Figure 2).

The temperature dynamics in the algal pond results from the following heat balance:(12)$$\begin{array}{ccc} \rho_{w}VC_{p,w}\frac{dT_{p}}{dt} & = & Q_{ra,p}+Q_{ra,s}+Q_{ra,a}++Q_{h,evap}+Q_{conv}+Q_{h,in}\\ \; & \; & +Q_{h,out}+Q_{h,rain}-Q_{conv,cond_{w},pp} \end{array}$$
where $T_{p}$ is the pond temperature (K); $\rho_{w}$ is the density of pond water (kg m${}^{-3}$); $C_{p,w}$ is the specific heat capacity of the pond water (J kg${}^{-1}$ K${}^{-1}$); and *V* is the pond volume (m${}^{3}$). $Q_{ra,p}$ is the radiation from the pond surface (W); $Q_{ra,s}$ is the total (direct + diffuse) solar radiation (W); and $Q_{ra,a}$ is the radiation from the air of the pond (W). $Q_{h,evap}$ is the evaporation flux (W), $Q_{conv}$ is the convective flux at the pond surface (W); $Q_{h,in}$ is the heat flux associated with the influent water (W); $Q_{h,out}$ is the heat flux associated to the effluent water (W); and $Q_{h,rain}$ is the heat flux related to rain (W).

The additional term, compared to the original model [13], is $Q_{conv,cond_{w},pp}$, the conductive/convective expressions related to the heat exchange between the water in the pond and the material of the HRABP (here, polypropylene (pp)):(13)$$Q_{conv,cond_{w},pp}=h_{w,pp}\cdot \left(T_{p}-T_{pp}\right)\cdot S$$
where *S* is the pond surface $\left(m^{2}\right)$.

The parameter $h_{w,pp}$ (W m${}^{-2}$ K${}^{-1}$) is the heat transfer coefficient between the water in the pond (w) and the polypropylene, while $T_{pp}$ (K) is the temperature of the polypropylene.

The HRABP was raised from the ground, as shown in Figure 1. A heat balance on the polypropylene mass of the pond is therefore also carried out:(14)$$\begin{array}{ccc} \rho_{pp}V_{pp}C_{p,pp}\frac{dT_{pp}}{dt} & = & Q_{conv,cond_{w},pp}+Q_{conv,cond_{a},pp}+Q_{ra,pp}+Q_{ra,a,pp}+Q_{rad,g} \end{array}$$
where $Q_{ra,pp}$ is the pond radiation at the bottom of the reactor:(15)$$Q_{ra,pp}=-\epsilon_{pp}\cdot \sigma \cdot T_{pp}^{4}\cdot S$$
and $Q_{ra,a,pp}$ is the expression describing the air radiation at the bottom of the pond ($T_{a}$ (K) is the air temperature), on the polypropylene surface:(16)$$Q_{ra,a,pp}=\epsilon_{pp}\cdot \epsilon_{a}\cdot \sigma \cdot T_{a}^{4}\cdot S$$

The conductive/convective flux between air and the material of the pond is $Q_{conv,cond_{a},pp}$:(17)$$Q_{conv,cond_{a},pp}=h_{a,pp}\cdot \left(T_{a}-T_{pp}\right)\cdot S$$

The parameter $h_{a,pp}$ (W m${}^{-2}$ K${}^{-1}$) is the heat transfer coefficient between the air (a). Finally, $Q_{rad,g}$ is the radiation from the ground to the pond bottom:(18)$$Q_{rad,g}=\epsilon_{ground}\cdot \sigma \cdot T_{g}^{4}\cdot S$$

The temperature of the ground was computed from the air temperature, according to the work of Tsilingiridis et al. [22]:(19)$$T_{g}=1.197\cdot T_{a}-0.7776;$$

The parameters $\epsilon_{ground}$, $\epsilon_{pp}$, and $\epsilon_{a}$ are the emissivity of the ground, polypropylene, and air, respectively (-); σ is the Stefan–Boltzmann constant $\left(Wm^{-2}K^{-4}\right)$.

The evaporative and convective heat flux in the liquid medium have a marked effect on the pond temperature prediction, and they were implemented according to the Buckingham theorem, depending on the dimensionless numbers of Sherwood, Schmidt, and Reynolds. The Sherwood and Nusselt numbers were linearly interpolated for Reynolds numbers between 3 × 10${}^{5}$ and 5 × 10${}^{5}$ to ensure model calculations for Reynolds numbers in the transition from laminar to turbulent flow. For the calculation of the evaporative heat flux, a minimum wind speed (4 m s${}^{-1}$) was considered, as the equation for evaporative heat flux does not account for evaporation when there is no wind.

### 3.3. Exchanges with the Atmosphere

The CO${}_{2}$, O${}_{2}$ stripping/dissolution and NH${}_{3}$ stripping are also included, quantifying their rates through the k${}_{L}$*a* and their diffusion coefficients:(20)$$Q_{j}=k_{L}a_{j}\cdot {\left(\frac{D_{S_{j}}}{D_{S_{O_{2}}}}\right)}^{2}\cdot \left(S_{j,sat}\left(T_{p}\right)-S_{j}\right)$$

From the evaporation and rain heat fluxes, it is possible to evaluate the water level variation in the pond:(21)$$\frac{dh_{L}}{dt}=\frac{-m_{e}}{\rho_{w}}+q_{r}+\frac{q_{i}}{S}-\frac{Q_{out}}{S}$$
where $m_{e}$ is the evaporation rate (Kg s${}^{-1}$ m${}^{-2}$); $\rho_{w}$ is the water density (Kg m${}^{-3}$); $q_{r}$ is the rain rate (m${}^{3}$ m${}^{-2}$ s${}^{-1}$); and $q_{i}$ is the inflow rate (m${}^{3}$ s${}^{-1}$).

### 3.4. Chemical Submodel

The chemical submodel consists of predicting pH ($\left(H^{+}\right)$ ions concentration), but also computing the fraction of the different dissociated compounds that are in equilibrium, depending on the pH. In Appendix B, we recall the principle of the implementation to highlight the temperature dependence of this sub-model, by detailing the set of algebraic equations to be solved.

The strategy for resolving the pH subsystem was inspired by the work of Rosén et al. [23]. The idea is to transform the initial problem of resolution of a set of algebraic equations resulting from mass balance and chemical dissociation into the solution of a differential system (see Appendix B).

## 4. Results and Discussion

### 4.1. Validating the Temperature Model

The heat transfer model was simulated using the nearby meteorological data provided by ARPA Lombardia regional meteo database. The temperature predictions are accurate, as can be seen in Figure 3 and in Table 1. The model tends to slightly overestimate the temperature in the warmest periods and to underestimate it in the coldest ones. The temperature predictions stay precise, with an error standard deviation of 1.6 ${}^{{}^{\circ}}$C, computed on the difference between the measured and the simulated temperature vectors, and a *p*-value below $10^{-6}$.

Model predictions could be further improved by tuning some of the heat transfer model parameters, or by using a dedicated meteorological station providing a more accurate estimation of the environmental conditions at the pond location. In fact, the weather dataset used for running simulations (i.e., air temperature, relative humidity, wind speed, and rain rate) were coming from the meteorological station at a distance of 10 km from where the reactor was located.

### 4.2. Full Predictive ALBA Model

In order to evaluate the overall model prediction capability when the physical and the biological models are coupled, the predicted temperature, instead of the one measured with the probe, was used to simulate the dynamics of the biological process over the 6-month period. The most relevant model state variables are reported in Figure 4, Figure 5, Figure 6 and Figure 7, and compared with the experimental measurements, their standard deviations, and the error bounds for model predictions.

The accuracy of the heat transfer model results in predictions that are almost unchanged compared to the ones using the measured temperature (see Table 1). When comparing the predicted measurements of the initial model with measured temperature and the new one with predicted temperature, the $r^{2}$ is always larger than 0.98 for 2000 simulated points, and the predicted variables stay within an interval of 3% compared to the values simulated using the measured temperature.

### 4.3. Process Configuration and Depth

The model was then used to evaluate the consequence on two configurations on the process productivity, in terms of algae production or on TAN removal rate.

An alternative HRABP process was considered for comparison purpose. It consisted in a more classic set up of the raceway, with a contact on the ground (see Figure 8). Another heat transfer model (see details in Appendix C) was developed for this configuration [24], also extending the one of Béchet et al. [13]. The full predictive model was adapted to this particular case, and was also validated over more than one year with a HRABP lying on the ground (see details in Appendix A). Water temperature was also accurately forecast [24] (see Appendix D). The connection with the ground strongly increases the thermal inertia, and the objective of the study is to determine how much it impacts the process efficiency.

The study was also carried out considering three different water depths ($h_{L}$): 0.232 m, 0.116 m and 0.077 m, respectively. The first value is the nominal liquid depth that was applied during the experimental campaign described in Section 2, and for which the model was validated. The two other depth values, which are also typical of industrial scale raceways, result from the scenario where 2 raceways (respectively 3) are used to treat the same inflow rate. The total volume and HRT for each scenario are the same, but depending on the scenarios (i) one raceway of depth 0.232 m is used, (ii) 2 raceways of the same area but with depth 0.116 m are used and (iii) 3 raceways of depth 0.077 m are operated.

It must be stressed that depth values represent the maximum liquid level reachable in the reactor. In fact, since the liquid depth is an output of the thermal balance, its level profile always oscillates because of the evaporation and rain rate contributions.

Looking at the medium temperature ranges in each situation, it turns out that the temperature oscillations when the pond is not connected to the ground is of much larger amplitude. The average number of hours per day when the temperature is higher than 35 ${}^{{}^{\circ}}$C was computed, according to season (Table 2). This temperature was identified by Bechét et al. [25] as a lethal temperature for many species, with dramatic consequences in terms of viability [26]. In summer, on average 4 h per day a temperature higher than 35 ${}^{{}^{\circ}}$C can be reached, for the thinnest process raised above the ground. This daily average is reduced to 2 h for the raceway on the ground. It is lower for liquid depth of 0.2 m, but it is still larger than 1 h per day. A closer look at extreme events (see Figure 9) shows that, for the hottest days, the medium stays more than 7 h above the critical threshold, which is likely to lead to the culture crash. A side effect of these high temperature is that evaporation losses are much higher in the raised HRABP, with an average increase by 20%.

Although our biological model cannot predict the impact of these high-temperature events, it is very likely that the ecosystem is degraded during these periods. This type of pilot therefore imposes unfavorable conditions for algal growth during the hottest months, at least under Mediterranean climates.

### 4.4. Ecosystem Composition

Experimental measurement of the various algal and bacterial biomasses is challenging. In the work of Ibekwe et al. [27], DNA extracts were used to assess the microbial community structure from bacterial 16S rRNA gene analysis (or 18S rRNA for algae) in an algae-bacteria raceway pond. These approaches are intensive and can only be carried out on a very limited number of samples. Above all, they provide an estimate of the number of operational taxonomic units (OTUs), but do not give a picture in terms of biomasses. A validated model such as the ALBA model can estimate the ecosystem composition over time.

Figure 10 presents the ecosystem composition estimated by the model, according to seasons and to the various tested conditions.

Microalgae dominate the system in all the seasons and raceway configurations. The average proportion of heterotrophic and nitrifying bacteria on the total biomass increases in autumn (7.6–8.8% for heterotrophs and 11.9–13.3% for nitrifiers), for both raceways layouts, but remains marginal compared to algae concentration. The fraction of nitrifiers stays marginal along the various scenarios (1–1.4% in spring, 1.1–2.7 in summer, and 5.3–13.3% in autumn); however, it is worth noting that they are processing more than 70% of the influent ammoniacal nitrogen. Heterotrophic bacteria are higher in concentration in spring and summer (17.8–22.1 and 20.2–37.3 gCOD m${}^{-3}$, respectively) for both raceway layouts compared to nitrifiers, while in autumn, these two populations are more balanced. As expected, for lower depths, algae concentration increases, resulting from the fact that algae are favored in light absorption.

In general, the bacterial concentration is only marginally affected by depth. Nitrifying bacteria are indirectly affected by the competition with microalgae for inorganic carbon, and their concentration is thus higher for the higher liquid depths, associated to lower microalgal concentrations. The two raceway configurations do not massively affect the biomass distribution. A model that would more accurately account for the consequences of the extreme temperature events on mortality, according to the different populations, might modify these conclusions. For this, additional experiments to quantify how the high-temperature periods affect bacterial mortality are necessary.

### 4.5. Algal Productivity and Ammonium Removal Rate

Looking at the biological performances for the two different configurations, it turns out that rather similar values for the daily average are obtained (see Figure 11), and the results are first of all influenced by water depth.

Algae productivity ranges between 3.6 and 9.3 (gDW m${}^{-2}$ d${}^{-1}$), in line with the values recorded in other raceways pond applications [28,29], with a maximum productivity in summer. The rather low considered HRT explains these modest performances, which would have been higher for lower HRT. Algae concentration increases when operating with lower depth, but not in the same order of magnitude: when depth is divided by a factor of 2, algae concentration is increased by 30% in spring and by 110% in summer, explaining why the algal productivity is not eventually enhanced. This is mainly due to: (i) a stronger influence of the evaporation rate at lower water levels; and (ii) a much stronger competition for CO${}_{2}$ with nitrifiers at higher algal density, reducing the algal growth rate [10]. The competition for inorganic carbon appears, even if pH is regulated at 7.5, due to a strong alkalinity consumption from nitrifying activity. Adding alkalinity in the influent has been shown to be an effective strategy to reduce and avoid this competition. In fact, with this approach, algae have the possibility to grow at their maximum rate, avoiding limiting conditions on inorganic carbon and thus increasing productivity. This is even beneficial for the environment, since inorganic carbon shortage has been shown to be associated with N${}_{2}$O emission by nitrifiers [30].

The TAN removal rate follows a similar trend, also driven by the inorganic carbon competition in the HRABP. In this case, since most of the ammonium is removed by the nitrifiers, reducing depth decreases the nitrogen flux treated by nitrifiers in similar proportion (the volumetric TAN removal rate being much less affected). The consequence is that the influent nitrogen to algae conversion rate considerably increases when reducing the liquid depth, since a higher proportion of nitrogen is processed by algae.

The similarity between the process on the ground and the one that is raised shows that mainly average temperature matters, and the daily average values do not significantly differ. However, looking closer at the standard deviation, it turns out that the predicted productivity varies widely. Temperature with thinner systems fluctuates strongly, leading to both variation in growth rate and in algal concentration. The large changes in concentration are also due to evaporation for the lowest depth. This explains why the standard deviation of algae concentration is more than 4 times higher at lower depth compared to its nominal value in summer. This is much less marked for the other seasons, when a factor only 2 is involved.

Keeping in mind that thinner system, and the raised process experiment with higher temperature ranges, with frequent excursions above the critical temperature, it is clear that reducing depth at constant HRT does not improve the process efficiency, and there is a risk of process crash due to high cell mortality.

A model to better represent the mortality at high temperature must be included in the biological model to propose a more accurate vision of the underlying processes.

### 4.6. Choice of the Most Efficient Solutions

The different system configurations, in terms of overall productivity and TAN removal rate, were compared (Figure 12). Reducing the depth by a factor 2 or 3 doubles or triples the number of raceways needed to process the same flux. The average TAN removal, installing two or three raceways, increases only by 3.4% in spring, 3.7 and 4.8% in summer, and 2.6 and 4.5% in autumn, respectively, for the raceway lying on the ground. Looking at the raceway configuration that is raised up, comparable low gains in TAN removal rate are reached in all the seasons (3.1 and 3.4% in spring, 3.3 and 4.2% in summer, and 2.7 and 4.8% in autumn, respectively). The moderate increase in total TAN removal rate is most probably not compensated by the higher capital and operational costs.

On the contrary, if the objective is algae production, then using two or three raceways instead of one increases the total biomass production by 21.4 and 30.3% in spring, 64.3 and 94.0% in summer, and 80.7 and 132% in autumn, respectively, for the raceway lying on the ground. For the raceway configuration that is raised up, the percentage of increased algae productivity is in the same range (18.2 and 27.6% in spring, 57.7 and 91.9% in summer, and 69.1 and 113.9% in autumn, respectively). This strong production enhancement probably justifies that three raceways with lower depth are used.

## 5. Conclusions

The fully predictive model for simulating outdoor algae-bacteria processes includes a biological, a chemical, and a thermal submodel. The biological predictions, when temperature is also forecast, were revealed to be very accurate. Temperature dynamics follow seasonal variability according to reactor configurations and liquid depth, playing an important role in the biological and chemical submodels.

For lower depths or for raceways not directly lying on the ground, the resulting over-warming does not significantly alter productivity, since the daily average temperature is only marginally affected. However, when the model predicts a large deviation of temperatures during the warmest seasons, it means that algae are exposed to lethal conditions, and there is a strong risk of culture crash.

This work definitely motivates a model-based thermal design strategy in order to reduce the duration of extreme-temperature events and their dramatic consequences on the process integrity.

## Figures and Tables

**Figure 1 microorganisms-10-01515-f001:**
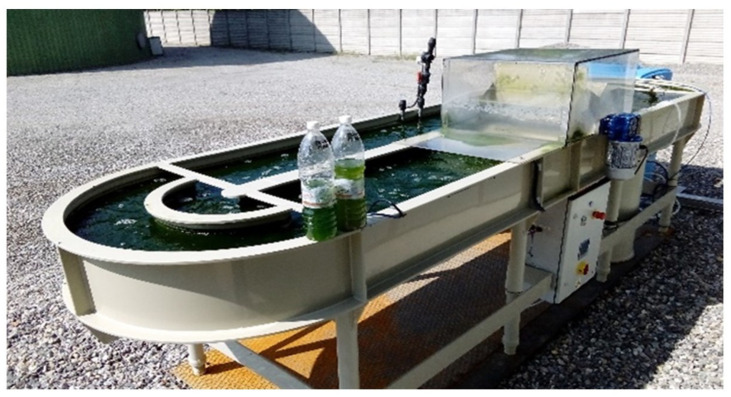
Pilot-scale HRABP located in a piggery farm in the north of Italy (Casaletto di Sopra, Cremona, Italy).

**Figure 2 microorganisms-10-01515-f002:**
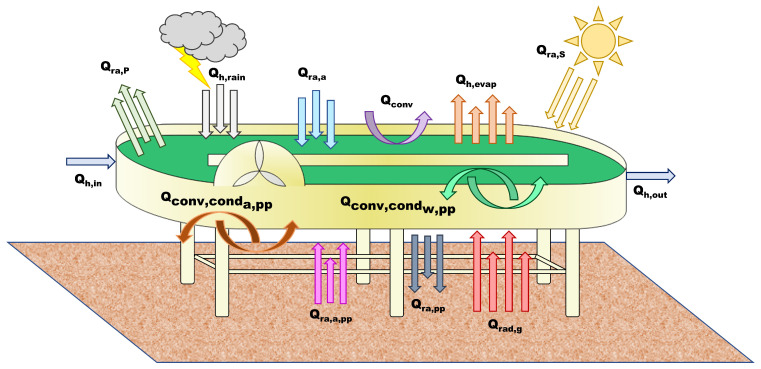
Heat fluxes considered in the radiative transfer model.

**Figure 3 microorganisms-10-01515-f003:**
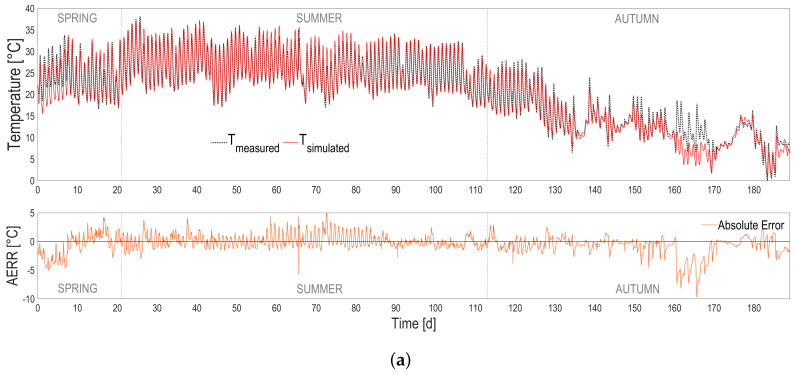
(**a**) Temperature predictions (red continuous line) compared with measurements (black dotted line) according to seasons. Absolute prediction error (orange continuous line) is also shown. (**b**) Temperature predictions vs. measurements (red dots) and histogram of residuals (blue bars).

**Figure 4 microorganisms-10-01515-f004:**
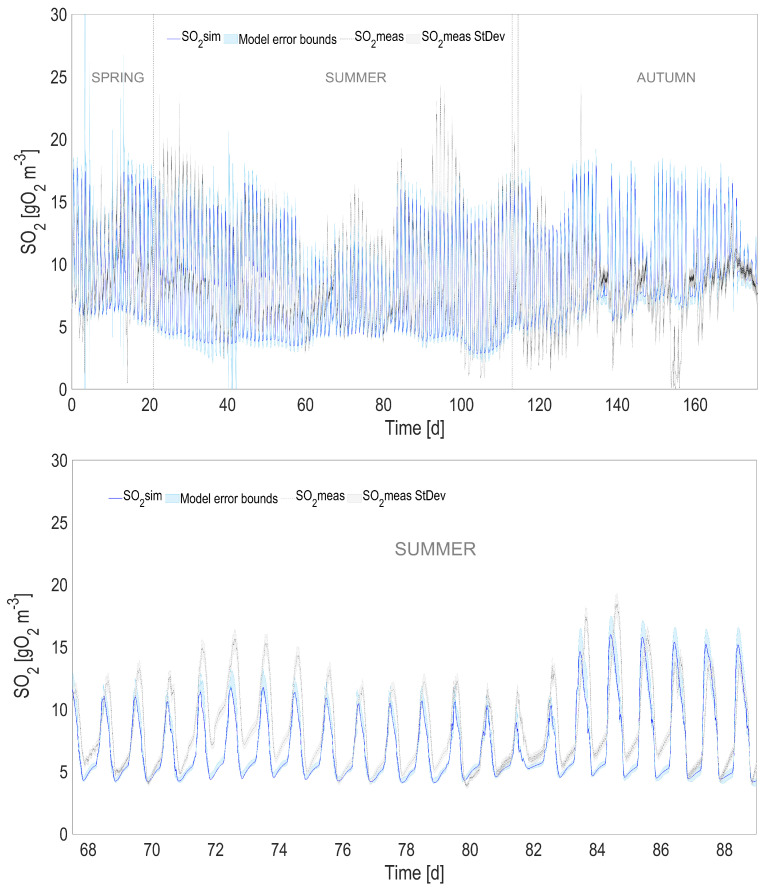
Full prediction capability of the coupled heat transfer and ALBA models for $O_{2}$, compared with simulations using measured temperature (with a zoom in the lower figure). The shaded areas represent the 95% confidence intervals of model predictions.

**Figure 5 microorganisms-10-01515-f005:**
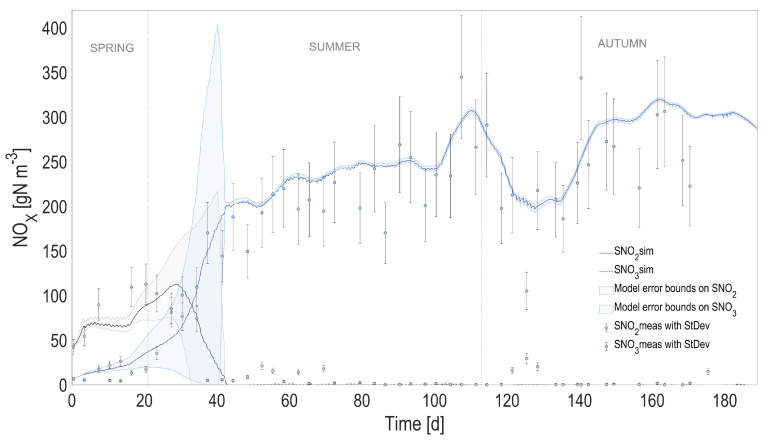
Full prediction capability of the coupled heat transfer and ALBA models. Comparison between full predictions (with simulated temperature) and measurements, for nitrite ($S_{NO_{2}}$) and nitrate ($S_{NO_{3}}$). Shaded areas on model predictions represent the related 95% confidence intervals.

**Figure 6 microorganisms-10-01515-f006:**
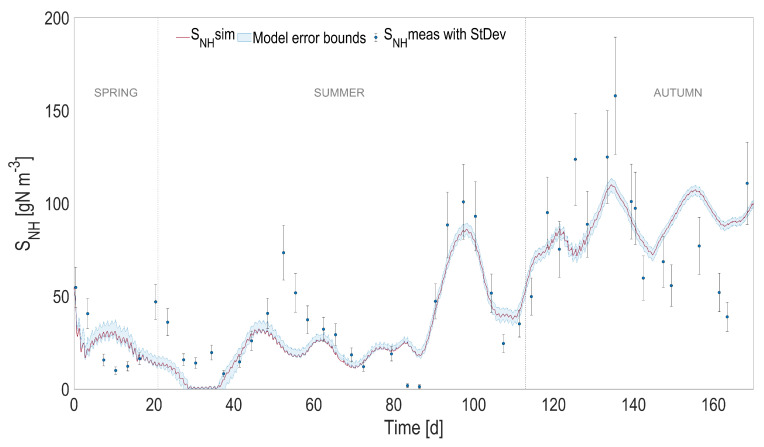
Full prediction capability of the coupled heat transfer and ALBA models. Comparison between full predictions (with simulated temperature) and measurements, for ammoniacal nitrogen ($S_{NH_{4}}$). Shaded areas on model predictions represent the related 95% confidence intervals.

**Figure 7 microorganisms-10-01515-f007:**
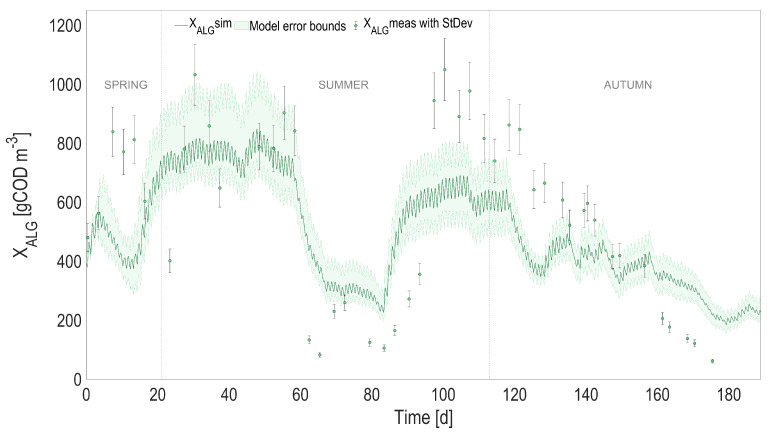
Full prediction capability of the coupled heat transfer and ALBA models. Comparison between full predictions (with simulated temperature) and measurements, for algal biomass ($X_{ALG}$). Shaded areas on model predictions represent the related 95% confidence intervals.

**Figure 8 microorganisms-10-01515-f008:**
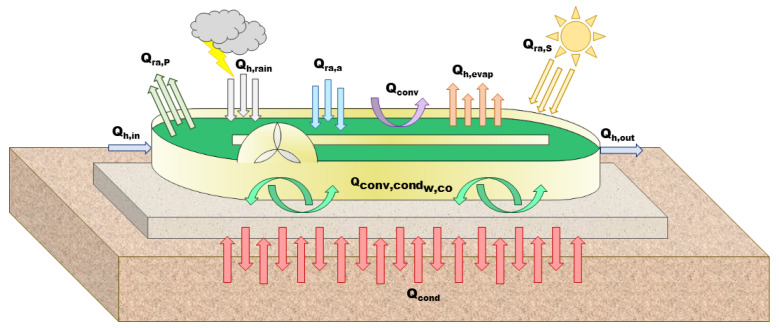
Scheme of the standard HRABP lying on the ground, with the heat fluxes considered in the radiative transfer model.

**Figure 9 microorganisms-10-01515-f009:**
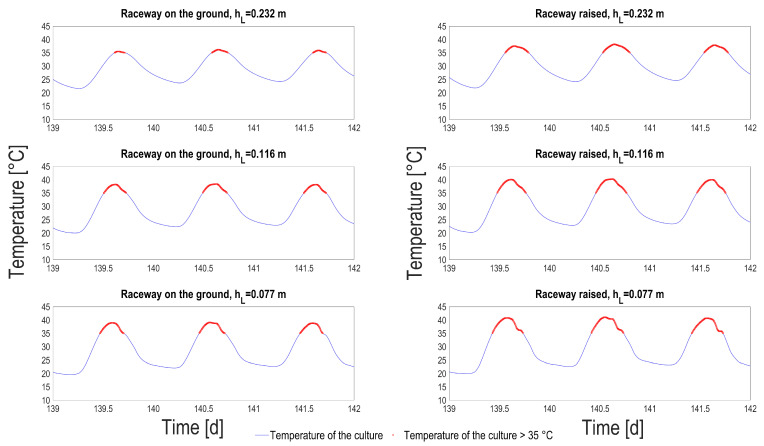
Pond temperature predictions (blue line) and simulated temperature higher than 35 ${}^{{}^{\circ}}$C (red dots), according to the liquid depth tested ($h_{L}$) and for both the raceway configurations: on the ground (**left** column); raised (**right** column).

**Figure 10 microorganisms-10-01515-f010:**
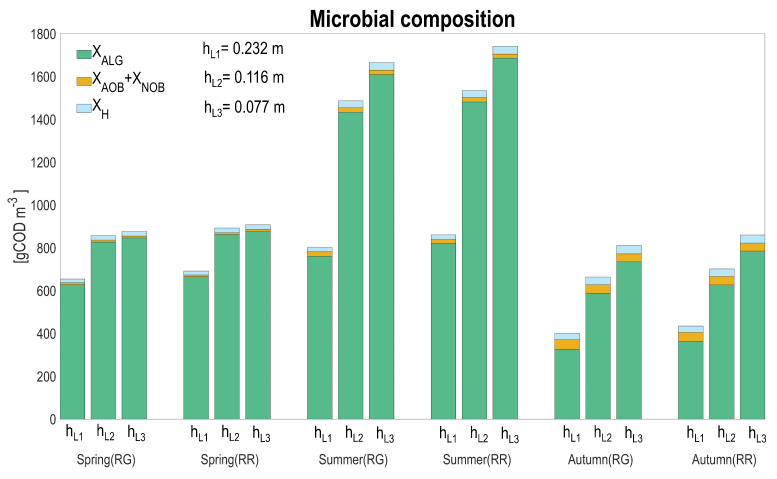
Seasonal average for biomass composition: algae ($X_{ALG}$), ammonium oxidizing bacteria ($X_{AOB}$), nitrite oxidizing bacteria ($X_{NOB}$), heterotrophic bacteria ($X_{H}$), for the two raceway configurations (RG: raceway on the ground, RR: raceway raised) and according to the three water depths.

**Figure 11 microorganisms-10-01515-f011:**
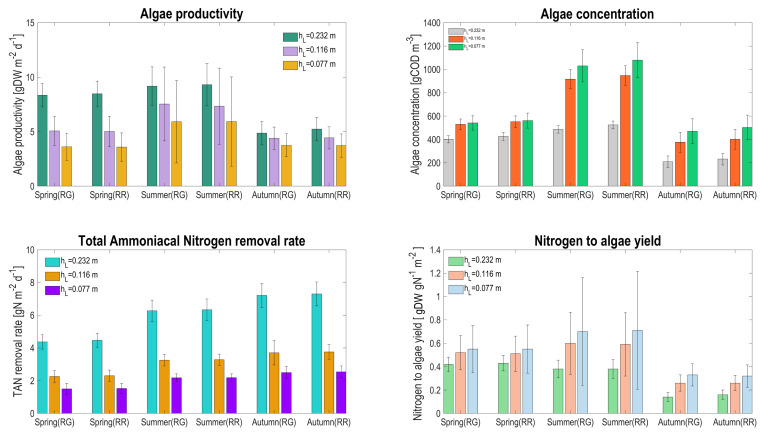
Seasonal average and the associated standard deviation for algal productivity, algae concentration, TAN removal rate, and nitrogen conversion rate, for the two raceway configurations (RG: raceway on the ground, RR: raceway raised) and according to the three water depths.

**Figure 12 microorganisms-10-01515-f012:**
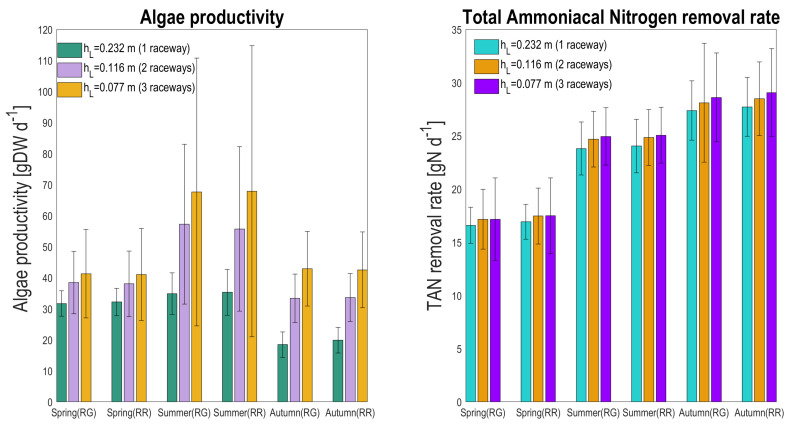
Seasonal average total algal production (gDW d${}^{-1}$) and total TAN removal rate (gN d${}^{-1}$) for the two raceway configurations (RG: raceway on the ground, RR: raceway raised), and depending on the number of raceways to process the same inflow rate (with the same HRT). Scenario with one, two, or three raceways are associated to depths $h_{L}$ of 0.232, 0.116, and 0.077 m, respectively.

**Table 1 microorganisms-10-01515-t001:** Comparison between simulated variables using temperature measurement and predictions from the full predictive model. (*): for temperature, the comparison is between the measured and simulated temperatures.

State	Err.St.Dev.	Abs.Rel.Err.	Coeff.Var.	$R^{2}$
Variable	(av)	(av %)	(-)	(-)
$S_{NH}$	0.9165	3.0531	0.0170	0.9994
$S_{NO_{2}}$	1.7102	7.7069	0.1004	0.9982
$S_{NO_{3}}$	1.8548	1.6769	0.0087	0.9997
$X_{ALG}$	14.6689	2.5129	0.0291	0.9940
TSS	10.3061	1.7773	0.0228	0.9952
CODs	0.6988	0.3022	0.0023	0.9998
$S_{O_{2}}$	0.4511	2.8032	0.0533	0.9844
pH	0.0531	0.4669	0.0080	0.9912
*T* *	1.6178	7.4652	0.0774	0.9632

**Table 2 microorganisms-10-01515-t002:** Average number of hours per day where the medium temperature is larger than the critical temperature (35 ${}^{{}^{\circ}}$C), for the two configurations (RG: raceway lying on the ground, RR: raceway raised above the ground) and for three water depths (h${}_{L}$).

h${}_{L}$	Spring		Summer		Autumn	
[m]	RG	RR	RG	RR	RG	RR
0.232	0.00	0.00	0.19	1.18	0	0
0.116	0.33	1.00	1.44	3.21	0	0
0.077	0.72	1.33	2.08	3.99	0	0

## Data Availability

Data supporting the reported results con be found in the Appendix A, Appendix B, Appendix C, Appendix D, Appendix E, Appendix F, Appendix G,Appendix H and Appendix I.

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
