# Peer review of "How Heat Transfer Indirectly Affects Performance of Algae-Bacteria Raceways"

_microorganisms, 2022, doi:10.3390/microorganisms10081515_

Round 1

Reviewer 1 Report

After considering the changes the authors have made in the manuscript, publication in its current form is recommended.

Author Response

We are grateful to Reviewer #1 for his/her positive assessment of our work.

Reviewer 2 Report

This manuscript improved a heat transfer model to compute algal productivity as a function of temperature oscillations. The manuscript had some innovation and significance, but also had shortcomings. Therefore, it was suggested to be published after major revision. Some examples are presented below.

1.     For the title, please be clear about the meaning of productivity. Does it mean algal biomass productivity?

2.     In the abstract and conclusion, please add the necessary data to show the result and accuracy of model.

3.     In the introduction, you should list the latest findings and research status about the heat transfer model of algae-bacteria raceways so as to more clearly present your contributions.

4.     What are the main reasons for borrowing this model and improving it?  What is the significance of this improvement? The author should reframe the language to highlight innovation in the introduction.

5.     In line 208, there need 2 spaces before “The two additional terms……”.

6.     For figure 3, as the author said about “The model tends to slightly overestimate the temperature in the warmest periods and to underestimate it in the coldest ones.”, what about the accuracy of temperature predictions in winter? Including other data, please explain the reasons for not using winter’s data in the manuscript. Moreover, the coordinate title of the chart should begin with a capital letter, consistent with other charts.

7.     In Figure 9, the legend font is too small to read clearly. Please adjust the font size.

8.     For “4. Ecosystem composition”, the methods for measuring microbial composition are not described.

9.     In line 315, there need 2 spaces before “In general”.

Author Response

This manuscript improved a heat transfer model to compute algal productivity as a function of temperature oscillations. The manuscript had some innovation and significance, but also had shortcomings. Therefore, it was suggested to be published after major revision. Some examples are presented below.

  1. For the title, please be clear about the meaning of productivity. Does it mean algal biomass productivity?

Our work focuses on two criteria: algal productivity and nitrogen removal rate. Changing "productivity" in the title into "algal productivity" would have shadowed the other criterion. In the end, we changed the title using "performance" instead of "productivity".

  1. In the abstract and conclusion, please add the necessary data to show the result and accuracy of model.

We have added a sentence in the abstract and in the conclusion to highlight the model accuracy.

"A heat transfer model was developed and  it was able to accurately predict the temperature during the year (with a standard error of 1.5° C). The full predictive model,  using the temperature predictions, degraded the model predictive performances by less than 3%. NO_2 predictions were affected by +/- 7 %, highlighting the sensitivity of nitrification to temperature.

  1. In the introduction, you should list the latest findings and research status about the heat transfer model of algae-bacteria raceways so as to more clearly present your contributions.

We more extensively and explicitely refered to the existing heat transfer models:

"Several heat transfer models have been proposed for algal raceways. The most popular one was developed in Béchet et al. (2011,2018). A simplified model was proposed by Slegers et al. (2013). A combination of these two approaches was then used in Rodríguez-Miranda (2021),  for a raceway pond. "

  1. What are the main reasons for borrowing this model and improving it?  What is the significance of this improvement? The author should reframe the language to highlight innovationin the introduction.

We now better explain the purpose of our work in the introduction:

"In this paper, we are focusing on a different configuration, where the raceway is not directly in contact with the ground. For this specific configuration, there is not heat transfer model available. "

  1. In line 208, there need 2 spaces before “The two additional terms……”.

Thank your for this remark, it was corrected.

  1. For figure 3, as the author said about “The model tends to slightly overestimate the temperature in the warmest periods and to underestimate it in the coldest ones.”, what about the accuracy of temperature predictions in winter? Including other data, please explain the reasons for not using winter’s data in the manuscript.

Unfortunately, the experiment stopped before winter, and these data are not available. The experimental campaign lasted 6 months (from 31/05/2016 to 06/12/2016, as also reported in Pizzera et al., 2019).  We agree with Reviewer #2 that such period deserves a specific treatment. This will be the subject of another publication.

  1. Moreover, the coordinate title of the chart should begin with a capital letter, consistent with other charts.

Thank your for this remark, it was corrected.

  1. In Figure 9, the legend font is too small to read clearly. Please adjust the font size.

Figure 9. was upgraded.

  1. For “4. Ecosystem composition”, the methods for measuring microbial composition are not described.

This paragraph is referring to the ability of the model to extrapolate a picture of the microbial composition. Experimental measurements of these quantities are challenging, as we now explain in the introduction of this section:

"Experimental measurement of the various algal and bacterial biomasses is challenging. In Ibekwe et al. (2017), DNA extracts were used to assess the microbial  community structure from bacterial 16S rRNA gene analysis (or 18S rRNA for algae) in an algae-bacteria raceway pond. These approaches are heavy and can only be carried out on a very limited number of samples. Above all, they provide an estimate of the number of operational taxonomic units (OTUs), but do not give a picture in terms of biomasses. A validated model such as the  ALBA model can reveal precious to estimate the ecosystem composition along time."

  1. In line 315, there need 2 spaces before “In general”.

This has been corrected

Reviewer 3 Report

Microorganisms-1744508-peer-review-v1

The present study updated the previous ALBA model by employing a physical model for predicting temperature evolution and developing a heat transfer model. It is quite useful and of significance for the field. The writing of the manuscript should be further improved from the abstract to the conclusion, and some minor revisions should be considered before it can be published.

*Abstract: Please reduce the length of the abstract and make it concise.

*Line 190: Please revise the expression of the citation [13] since the format is not suitable here. Please also revise the whole manuscript.

*4.1. Validating the temperature model: Temperature is important for the heat transfer model. The authors used the local meteorological data to simulate the changes in temperature. How does this model serve for temperature prediction? Since the temperature is an overall parameter that is affected by many factors. The trend of temperature could be varied year by year. So how this is solved in the model?

*Figure 3. The simulated and measured values are not easy to identify in the figure, please revise to make them clear.

*Conclusion: This section needs to be reduced. 

Author Response

The present study updated the previous ALBA model by employing a physical model for predicting temperature evolution and developing a heat transfer model. It is quite useful and of significance for the field. The writing of the manuscript should be further improved from the abstract to the conclusion, and some minor revisions should be considered before it can be published.

We are grateful to Reviewer #3 for his/her positive evaluation of our work.

*Abstract: Please reduce the length of the abstract and make it concise.

The length of the abstract was reduced.

*Line 190: Please revise the expression of the citation [13] since the format is not suitable here. Please also revise the whole manuscript.

We are sorry for the incorrect format of the citation. The expression for the citation [13] was corrected and the whole manuscript was revised.

*4.1. Validating the temperature model: Temperature is important for the heat transfer model. The authors used the local meteorological data to simulate the changes in temperature. How does this model serve for temperature prediction? Since the temperature is an overall parameter that is affected by many factors. The trend of temperature could be varied year by year. So how this is solved in the model?

The meteorological data (i.e. air temperature, relative humidity, wind speed and rain rate) are available from nearby meteo stations.  The temperature model for predicting the temperature in the reactor from the surrounding meteorology, uses these data, which are strongly dependent on the year, the season and the location. If another year needs to be simulated, the corresponding data must be downloaded first, and the temperature in the reactor will differ from the one simulated during the initial year.

*Figure 3. The simulated and measured values are not easy to identify in the figure, please revise to make them clear.

We modified the graphs in order to make it clearer the difference between measurements and simulation results. We also added a graph where the absolute prediction error appears.

*Conclusion: This section needs to be reduced. 

We reduced the length of the conclusion section.

Reviewer 4 Report

The experimental design has covered sufficient aspects of the fully predictive model for simulating outdoor . Algae-Bacteria process (including biological, chemical and thermal submodels). The results were reasonable. I would like to recommend the publication of this study in the Microorganisms if the following improvement could be made.

1. The authors should explain the novelty in this study in the Section Introduction. How is it different from previous studies?

Author Response

We are grateful to Reviewer #4 for his/her positive evaluation of our work.

  1. The authors should explain the novelty in this study in the Section Introduction. How is it different from previous studies?

We made it clearer what is the innovative contribution of our work.

"In this paper, we are focusing on a different configuration, where the raceway is not directly in contact with the ground. For this specific configuration, there is not heat transfer model available. "

and later

"It is important to stress that there are many pilot-scale raceway which are  not in direct contact with the ground, thus their thermal inertia and the associated temperature dynamics differ from the one taking place at industrial scale in raceways lying on the ground.
In fact, different reactor configurations are subjected to different heat-transfer contributions and balances, resulting in a different temperature inside the reactor. For instance, when the raceway is lying on the ground the conduction from the ground mitigates its temperature, while when the reactor is raised-up from the ground, the radiation from the ground heats the pond bottom.

With this new fully predictive model version, it became possible to compare the performances of a raised raceway with those of an industrial configuration, where the raceway is made of concrete and lies on the ground. Different water depths were also tested, and they  corresponded to different thermal inertia. "

Round 2

Reviewer 2 Report

The author has made some modifications according to the suggestions, but there are still some minor shortcomings in the manuscript. Some examples are presented below.

1.    It is not recommended to use grey font for “Spring, Summer and Autumn” in the figures, because it is not clear enough.

2.    Pay attention to the subheading format. Does “3.2. Predicting pond temperature and water level” only include “3.2.1. Exchanges with the atmosphere”?

3.    There needs a blanket in between “RR:” and “raceway” in Table 2.

4.    Please express SNO2 or SNO2 uniformly. “SO2, SNH, SNO3, SNO2” are in the same situation.

Author Response

1. It is not recommended to use grey font for “Spring, Summer and Autumn” in the figures, because it is not clear enough.

The grey font for "Spring, Summer, Autumn" was changed in black color.

2. Pay attention to the subheading format. Does “3.2. Predicting pond temperature and water level” only include “3.2.1. Exchanges with the atmosphere”?

The subheading format of the sections “Predicting pond temperature and water level” and "Exchanges with the atmosphere” was corrected.

3. There needs a blanket in between “RR:” and “raceway” in Table 2.

A blank  between "RR" and "raceway" was added in Table 2.

4. Please express SNO2 or SNOuniformly. “SO2, SNH, SNO3, SNO2” are in the same situation.

The names of the variables SNO2, SO2, SNH, SNO3 have been standardized both in the graphs and in the manuscript.

This manuscript is a resubmission of an earlier submission. The following is a list of the peer review reports and author responses from that submission.

Round 1

Reviewer 1 Report

The topic of this article is very interesting and meaningful, however, the following issues should be addressed before it can be accepted.

  1. The background introduction in the abstract part is too long, and the main research content, important findings, and innovations of this paper are not clarified.
  2. Materials and methods are not written in detail enough, such as reactor operating conditions, water quality measurement methods, statistical methods, etc. are missing.
  3. The conclusion part is recommended to be rewritten, the content is not conclusive, and it is not recommended to cite literature in this part.

Reviewer 2 Report

This study delineated the Algae-bacteria (ALBA) model and coupled it with a physical model that predicted temperature evolution in the HRABP. The authors developed a heat transfer model which can predict the temperature. And then the refined model was used to analyse algal productivity and ammonium removal rate in two different configurations. However, the data analysis is not comprehensive and lacks the focus. The scientific significance of this study is not clear, and the contents of this study cannot support the title. The comments are as follows:

Abstract:

  1. “ALgae-BActeria” should be “Algae-bacteria”

Keywords:

  1. “HRABP” cannot be a keyword.

Introduction:

  1. Page 2, line 55: Only the data for half a year is recorded here. It is confusing that why pH, SO2, ….have been monitored for 1.5 years.2. “Since many studies are considering raised raceways, the question that we target is how far will an upscaled process, with a different temperature dynamics, differ from this behaviour.” It needs to be clearly explained to help readers understand.
  2. Page 2, line 57: “However, since such pilots are not in contact with the ground, their thermal inertia and thus the associated temperature dynamics differ from the one reached at industrial scale in raceways lying on the ground.” This sentence needs more references to support your point. How to ensure accurate prediction of temperature dynamics?
  3. “..with extreme values likely to impact the biological activity, especially in the hotter periods.” What does this sentence mean?

Materials and Methods

  1. Page 3, line 88: “the HRT was increased to 20 days”, Please try to justify such an adjustment. What is the basis for setting the hydraulic retention time (10 days or 20 days), microalgae biomass or the results of wastewater treatment?
  2. How to sample at three depths? Please describe the process in detail.
  3. How to pretreat the digestate? How to determine the algae yield and the chemical indexes of wastewater (pH, SO2, SNH…)?

Modelling approach

  1. Page 6, line 157: “succesfully” should be “successfully”, Please pay attention to the spelling of words

Results and discussion

  1. Page 9, line 210: “The temperature predictions are accurate, as it can be seen in figure 3 and in Table 1. The model tends to slightly overestimate the temperature in the hotter periods and to underestimate it in the cooler periods.” This conclusion cannot be drawn from Figure 3. It is recommended to change the color to make the data more clear.
  2. 4.1. Validating the temperature model: “…with a standard deviation of 1.6℃.” Is it the standard deviation at a certain point in time? The data analysis is confused.
  3. Page 14, line 258: “This means that, for certain hotter periods, it can last for more than 7 hours.” This statement requires more data and computational support.
  4. The English language should be improved to ensure that an international audience can clearly understand your text. For example, Page 14, line 258: “this competition due to alkalinity shortage appears even if (like here) pH is regulated at a low value.”

; Page 15, line 287: “The consequence is that the influent nitrogen conversion rate to algae, considerably increases when reducing depth: a higher proportion of nitrogen being processed by algae.” should be changed to “The consequence is that the influent nitrogen conversion rate to algae considerably increases when reducing depth: a higher proportion of nitrogen is processed by algae.”; “Hereafter we recall the main structure of the ALBA model, but we refer the reader to [10,12] for more details.”

Conclusions

  1. Page 16, line 311: “This is risk is less poignant for conventional concrete raceways lying on the ground?” is hard to understand. Perhaps this is a declarative sentence rather than a question, not a scientific expression.
  2. “We show, as already studied by [29] that water depth considerably affects the amplitude of the temperature fluctuations, putting the system in the regular risk of mortality due to over-warming.” not a proper way to use a reference in conclusion.
  3. “The model shows that despite this risk, productivity is mainly affected by the daily 313 average temperature.” It is a common sence.

Figures

  1. Figure 5: Please number the three figs separately.
  2. From Figure 5, the overall fitting effect is not good, especially the NOX at 40th day.
  3. Figure 3: The gray line is not clear. It is recommended to change the color

Tables

  1. It is hard to understand the data in all tables. Please add more illustration, unit of data and explanation of abbreviations.

Reviewer 3 Report

The authors used the validated ALgae-BActeria (ALBA) model to represent this process.

This model was validated with 623 days of outdoor measurements at two different locations and for the four seasons. However, until now, this model, like all other existing models, was not fully predictive as it required the measurement of water temperature to be run.

In this manuscript, the authors described (1) ALBA model (2) General kinetic model, (3) equation of Light impact, (4) temperature impact……. However, the aims of Microorganisms journal is to encourage scientists to publish their experimental and theoretical results and provides an advanced forum for studies related to prokaryotic and eukaryotic microorganisms, viruses and prions. In this manuscript, I find the limited data on “Microorganisms”, just showed the data about Algae productivity, concentrations..in Table 3. Furthermore, I did not find the information and method of productivity in the Methods section. In addition, I did not find the topic of ALgae-BActeria (ALBA) model in the Results section and showed the results in Figures. Since the result is important for this study, please add the analysis about ALgae-Bacteria.